# Insight on Incorporation of Essential Oils as Antimicrobial Substances in Biopolymer-Based Active Packaging

**DOI:** 10.3390/antibiotics12091473

**Published:** 2023-09-21

**Authors:** Ana Tomić, Olja Šovljanski, Tamara Erceg

**Affiliations:** Faculty of Technology Novi Sad, University of Novi Sad, Bulevar Cara Lazara 1, 21 000 Novi Sad, Serbia; anav@uns.ac.rs (A.T.); tamara.erceg@uns.ac.rs (T.E.)

**Keywords:** antimicrobial packaging system, antimicrobial incorporation, biopolymer matrix, food application, microbiological food safety

## Abstract

The increasing interest in microbiological food safety requires the development of sensitive and reliable analyses and technologies for preserving food products’ freshness and quality. Different types of packaging systems are one of the solutions for controlling microbiological activity in foods. During the last decades, the development of biopolymer-based active packaging with essential oil incorporation systems has resulted in technologies with exceptional application potential, primarily in the food industry. There is no doubt that this principle can facilitate food status monitoring, reduce food waste, extend the shelf life, improve the overall quality of food, or indicate a larger problem during the storage, production, and distribution of foodstuffs. On the other hand, most antimicrobial packaging systems are in the development phase, while the sensitivity, selectivity, complexity, and, above all, safety of these materials are just some of the essential questions that need to be answered before they can be widely used. The incorporation of essential oils as antimicrobial substances in biopolymer-based active packaging holds significant promise for enhancing food safety, extending shelf life, and offering more sustainable packaging solutions. While challenges exist, ongoing research and innovation in this field are likely to lead to the development of effective and environmentally friendly packaging systems with enhanced antimicrobial properties.

## 1. Introduction

Microbiological contamination is a significant concern in the food industry as it can lead to foodborne illnesses and the spoilage of products. There are several acute problems associated with microbiological contamination in the food industry: foodborne illness outbreaks, product recalls, regulatory compliance, consumer trust, economic loss, supply chain issues, etc. To address these acute problems, the food industry employs various strategies, including stringent sanitation practices, HACCP (Hazard Analysis of Critical Control Points) systems, regular testing and monitoring, and compliance with food safety regulations. Additionally, advances in food processing technologies as well as new packaging technologies are continuously being developed to mitigate microbiological contamination risks and improve food safety. On the other hand, the global production of plastics has gradually increased since plastics’ industrialization in the 1950s, reaching 368 million tons in 2019, with the expectation that it will reach 1.1 billion tons by 2050 [1]. The packaging industry is one of the main consumers of conventional plastic due to its good mechanical and barrier properties, lightweight, low-cost petrochemical building blocks, and established processing methods. Despite the numerous advantages, non-degradable plastic packaging has a negative impact on the environment and human health. Production of conventional plastic requires non-renewable fossil fuels as feedstock, resulting in the depletion of their reserves and increased emissions of greenhouse gases (GHG) [2]. Furthermore, the manufacturing, processing, and burning of plastic materials require high energy consumption, leading to the increased release of volatile organic compounds (VOCs) and harmful gases [3]. The enormous conventional plastic production, followed by the low level of recycling and high resistance to biodegradation, has resulted in the accumulation of a large quantity of plastic waste in nature. Over 45% of plastic waste comes from packaging, and even 90% of this amount comes from food and beverage packaging (wrappers, bags, containers, etc.). The slow degradation along with UV irradiation and abrasion results in the generation of microplastics, which end up in all ecosystems and living beings, causing different harmful effects. These facts have led to a growing interest in the development and manufacturing of biodegradable packaging materials obtained from renewable resources. Biopolymers, such as proteins and polysaccharides obtained by extraction from animal and plant materials, are degraded in the natural environment through chemical and biological processes, especially by the metabolism of microorganisms [4].

Beside plastic waste, food waste and spoilage are globally recognized as environmental and economic problems that need an emerging solution. Only a small percentage of food is composted, and the rest of the unused food is the largest compound of solid municipal waste, which degrades, releasing methane and contributing to greenhouse emissions. The food can be spoiled in physical, chemical, biochemical, and microbial ways during distribution, storage, and consumption. It is estimated that the amount of wasted food per year can feed one-eighth of the world’s population and answer the global challenge of meeting the increased need for food [5]. Packaging has a significant role in preventing and decreasing spoilage factors, acting as a barrier for water, gases, chemical, and microbiological agents, improving the safety of food, maintaining its quality, and extending the product’s shelf life. Perishableness of food and long-term distribution have imposed a need for the development of active packaging that is able to inhibit microbial growth due to the presence of antimicrobial compounds incorporated in a polymer matrix, which should be biodegradable according to environmental and ecological demands. Synthetic antimicrobial and antioxidant agents are used in the food industry, but some of them can be harmful to humans, causing resistance or disorder of the microbiological flora. Those facts, along with a demand for the utilization of renewable resources, have launched essential oils (EOs) into the focus of the scientific public as biobased compounds with pronounced antimicrobial and antioxidative effects, edibility, and harmlessness for human health that are suitable for application in food packaging [6,7].

Therefore, the development of active biodegradable packaging with incorporated bio-based antimicrobial agents such as essential oils can address urgent challenges on a global level, such as reducing plastic and food waste and their harmful effects on the environment. This review includes information about biopolymer-based structures used for packaging systems and different approaches for the incorporation of essential oils in targeted structures. Additionally, the emergence of microbiological contamination in the food industry, the influence of the packaging system on the inactivation of microbiological contamination, the migration of the antimicrobial substance into the packaging system, and the proposal for comprehensive determination of the antimicrobial potential of biopolymer-based active packaging with incorporated essential oils were summarized.

## 2. Emergence of Microbiological Contamination in the Food Industry

Diseases caused by the consumption of contaminated food, called foodborne diseases, represent a challenge to the entire system of food production and supply. The food supply line represents the processes of growing, harvesting, transporting, storing, and preparing food in an unhygienic environment, often without adequate environmental control measures. Therefore, this indispensable route represents an infection path for humans. Foodborne microorganisms often cause acute illness in humans but also significant economic losses during food production and processing. It should be noted that some groups of the human population are more at risk than others, such as children, pregnant women, elders with low immune status, persons with cancer, on chemotherapy, or/and infected with HIV [8]. There are thought to be more than 30 pathogens that are known to cause illness if present in food. Based on the World Health Organization (WHO) data, priority food contaminants are pathogen bacteria such as *Campylobacter* spp., *Escherichia coli* O157 (STEC and VTEC), *Listeria monocytogenes*, *Salmonella* spp. (non-typhoidal strains), *Salmonella enteritica* serotype Typhi, *Shigella* spp., and *Yersinia enterocolitica*. The mentioned bacteria are monitored in numerous countries and by a large number of health organizations for the prevention of foodborne diseases, determining the source and place of outbreaks, etc. [9].

In the last decade, the global food and beverage industry has seen significant growth. Following this trend, this industry is predicted to be worth more than 5700 billion dollars by the end of 2025 [10]. The turnover of the food and beverage industry in the reports of the European Union (EU) is estimated at 956.2 billion euros, whereby, within the framework of this community, over four million people are employed, with an over 20% increase in exports every year. Globalization and the increasing need for food introduce new risks into the system in terms of food safety, and contaminated food can be a source of risk that can spread over larger geographic areas [11]. Consumption of food containing an infectious dose of one of the mentioned pathogens affects every sixth resident in economically developed countries [12]. In addition to the mentioned bacterial contaminants, norovirus is at the top of the ladder. In addition to it, non-typhoidal species of *Salmonella* bacteria, *S. aureus*, *C. perfringens*, and *Campylobacter* spp. are found in five prominent foodborne pathogens [13]. Based on reports from 27 EU member states and four states that are not members of this community, a total of 5196 outbreaks of foodborne diseases were recorded in 2013. Most diseases are caused by the occurrence of salmonellosis, viral diseases, and the occurrence of bacterial toxins in food, while almost 30% of reported cases had an unknown source [14]. The most frequently contaminated foodstuffs are eggs, RTE (ready-to-eat) food, fish, and fish products. For example, the EU reported a total of 82,694 confirmed cases of salmonellosis in 2013, resulting in a reporting rate of 20.4 cases per 100,000 population, with 59 deaths. Furthermore, outbreaks of listeriosis were recorded in this community every year, while the number of confirmed VTEC infections in humans increased over the last ten years [15].

Microbiological contamination can enter the food chain in a production environment (farm, orchard, pond, etc.), process environment (slaughterhouse, factory, packaging plant, etc.), preparation environment (kitchen, food service, etc.), and/or water used in any production-process phase [16]. Although food spoilage can be attributed to many sources, such as enzymatic decomposition, physico-chemical reactions, damage during transport, and the influence of rodents and other pests, microbiological contamination has a decisive and very important contribution to food spoilage. Some of the most common non-pathogenic bacteria that cause food spoilage are *Pseudomonas* spp. and lactic acid bacteria (LAB), which are often involved in the spoilage of dairy and meat products. What distinguishes microbiological contamination in the food industry is that bacteria easily resist adverse environmental conditions, including high pH, low temperatures, and anaerobic environmental conditions [16].

### 2.1. Presence of Bacteria in Food

Bacteria occupy a very important place among the microorganisms found in food. This is not only because many species can be present in food but also because of their fast growth and ability to use nutrients, as well as their ability to grow in different conditions (wide range of temperature, presence of air, pH, water activity). Various indicators of fecal contaminants, such as *E. coli*, *Salmonella* spp., *L. monocytogenes*, *Vibrio cholerae*, and *Pseudomonas* spp., were isolated from water used in cases of agricultural production [10]. Also, farmed poultry and livestock are the primary sources of pathogenic bacteria that are often associated with human infections. Pathogenic bacteria can be “transferred” from infected or contaminated meat and become part of the food chain. Cross-contamination of foodborne pathogens in the retail sector is also a major public health issue, contributing to the increased risk of foodborne illness outbreaks. RTE food, such as meat delicacies, cheese, and other perishable ingredients, is often the cause of contamination with the bacteria *L. monocytogenes*. It is considered that bacteria are the main cause of spoilage in most food products, which causes a great economic loss in the process of food storage and preservation [12].

The presence of bacteria in food products is a matter of great importance. Bacterial motility, adaptability, sporogenicity, toxin production, and cell wall structure dictate the treatments involved in food preparation and processing. Although there are many ways to group bacteria, from the point of view of the food industry, the following division is the most important: non-pathogenic, conditionally pathogenic, and pathogenic bacteria, whereby the controversy that the mentioned groups may or may not be the cause of food spoilage must be added to the discussion [17]. Pathogenic microorganisms are the source of foodborne epidemics and cause everything from mild disease symptoms to fatal outcomes. Although a significant number of bacteria can be present in food, *E. coli*, *Salmonella* spp., and *Clostridium botulinum* stand out in particular and are extensively studied by both the medical and scientific public. An unavoidable detail is the unpredictable nature of bacterial contamination due to the type of food in which they are found, the environment and storage conditions, and the possibility of toxin production. Some bacteria are found in food products as secondary contamination, while others live only in certain food environments, such as *C. botulinum*, which has been a problem for the canned food industry for decades due to its anaerobicity and sporogenicity [18]. Another example of this is *L. monocytogenes*, which represents a major risk in RTE foods containing smoked meats and sausages [19]. Non-pathogenic bacteria, although less harmful to humans, indirectly affect the world economy by limiting access to food. For this reason, methods to preserve and slow down the rate of food spoilage have been a subject of interest since the beginning of the modern understanding of the concept of healthy and safe nutrition.

The most common foodborne bacteria, coupled with food sources and health risks, are summarized in Table 1.

### 2.2. Presence of Yeasts and Fungi in Food

Due to their ability to contaminate and degrade food products by producing extracellular enzymes, mycelial fungi and molds are of great importance in the microbiological contamination of food [20]. According to an estimate by the USDA Economic Research Service, about 96 billion kilograms of human food is discarded annually due to fungal contamination, with the largest share in the form of fresh fruits and vegetables (19.6%), milk and milk products (18.1%), grain products (15.2%), and sweeteners (12.4%). For example, in the fruit industry, post-harvest losses of 5 to 10% have been estimated when fungicides are used, and when fungicides are not used, losses amount to more than 50%. In the bakery industry, losses due to fungal contamination are up to 5%, which is over 23,000 tons of bread per year or 200 million pounds [21].

Among the fungi that appear as food contaminants, *Aspergillus* and *Penicillium* species stand out. In addition to them, *Mucor*, *Absidia*, and *Rhizopus* genera stand out. Apart from representatives of *Ascomycotina* and *Zygomycotina*, *xerophilic* fungi whose growth is characteristic under conditions of reduced water activity (aw) also occur as food contaminants. Representatives of this group of fungi are *Xeromyces* and *Vallemia* [21]. Representatives of the genus *Deuteromycotina* can also be food contaminants, with *Alternaria*, *Botrytis*, *Cladosporium*, and *Fusarium* species being the most prominent [17]. The most dominantly present foodborne yeasts and fungi are systematically present in Table 2.

Fungal contamination of food generally occurs first as contamination with spores that can be part of the sexual and asexual reproduction of fungi [22]. In favorable environmental conditions, spores break the resting phase, start germination, and create visible mycelia [20]. After a certain period, depending on the type of mold, food type, and storage conditions, the mycelium produces a series of conidia, and continuous development of the mold begins. This development is conditioned by environmental factors, but it is essentially impossible to prevent the spread of mold and food spoilage when favorable conditions for the initiation of the mold life cycle occur [22].

### 2.3. The Main Problem Related to the Presence of Microorganisms in Food

The presence of microorganisms in food is one of the main problems for the food industry, scientific community, and global economy. This problem appears to be a global issue because it represents a constant threat to health. As a consequence of the solution to this problem, the food industry must constantly develop and prepare procedures that minimize the occurrence and persistence of microbiological contamination in food [23]. This problem is also reflected in the fact that a large number of microorganisms can survive the effects of chemical agents allowed for use in food, as well as the physical and chemical procedures currently applied in the food industry to extend the shelf life [24]. Food safety depends to a large extent on appropriate government regulations, but also on the requirements and proper implementation of legal regulations, such as appropriate and constant training of people involved in food manipulation. The basic rules of food hygiene are essential, but they are still missing in many steps of food processing, such as the processes related to the slaughtering and processing of animal carcasses [25].

Modern methods for early detection of microbiological contamination include expensive genotyping and identifying the presence of microorganisms where possible and feasible in real-time [26]. However, a large number of food industries have introduced preventive measures for the occurrence of microbiological contamination. These procedures usually involve the use of antibiotics or alternative biocide compounds to suppress the growth of the microbiota. The long-term use of antibiotics leads to the emergence of bacterial resistance, creating a much bigger and longer-term problem for the food cycle in humans and animals. On the other hand, bacteria have the ability, after acquiring resistance, to suppress the effect of the antibiotic so that a critical intracellular concentration of the growth-inhibiting agent never accumulates [27].

In the last few decades, the resistance of microorganisms to antimicrobial agents has become a major environmental problem, and consequently, the number of broad-spectrum antimicrobial agents is decreasing. The acquired antimicrobial resistance, most microorganisms can transfer to new generations by simple, horizontal gene transfer. In addition, the abuse of antimicrobial agents, which is expressed in the veterinary and meat industries, has caused even specific antimicrobial agents to be no longer effective in the fight against pathogens in food. On the other hand, the pharmaceutical industry does not have a solution for the production of new antimicrobial agents, both due to the lack of sources and the high cost of production [27].

## 3. Biopolymer-Based Structures for Formation of Packaging Systems

Different materials have been used in food packaging, such as metal, glass, paper, and plastic. Environmental concerns and strivings to reduce the amount of solid waste, as well as consumers’ requirements for increasing food safety, have imposed a need for the development of biodegradable active packaging [28]. The differences between classical and biodegradable food packaging, along with the adventages and limitations, are summarized in Table 3.

Biopolymer-based structures for the encapsulation of essential oils can be divided into particles, fibers, and gels (Figure 1).

They usually have nanodimensions, which classify them as nanostructures. Different biopolymers have been utilized for the preparation of nanoparticles—vesicles with sizes up to 100 nm, intended for the encapsulation of essential oils in active packaging applications. Proteins and polysaccharides, individually or together (in the form of complexes), are used for obtaining nanoparticles. For the production of protein, polysaccharides, and protein/polysaccharides-based nanoparticles, different methods have been utilized, such as nanospray drying and complex coacervation. Different proteins and polysaccharides such as zein, gelatin, soy protein, whey protein, milk proteins, alginate, pectin, chitosan, pullulan, starch, cellulose, gellan gum, and maltodextrin have been successfully used for the formation of nanoparticles or nanocapsules intended for the incorporation of essential oils as antimicrobial agents in active packaging with the aim to achieve their compatibilization with hydrophilic biopolymer matrix and protect them from evaporation, thermal degradation, and oxidation during film processing and product storage [29,30]. The spray-drying technique enables the micro or nanoencapsulation of essential oils by dispersing them in an aqueous solution of film-forming biopolymer to obtain emulsion, which is pulled into the dryer through a nozzle providing small droplets. Improved thermal stability and controlled release of essential oils in active packaging formulations can be achieved by microencapsulation of essential oils by complex coacervation when two oppositely charged biopolymers make an electrostatic complex, which results in phase separation. As a result of neutralization, the dispersion is separated into two phases—one rich in biopolymer (coacervate) and the other with a low amount of biopolymer. This process implies the emulsification of essential oils in an aqueous solution of cationic and anionic biopolymers, adjustment of pH value, biopolymer ratio, concentration, and as a last step, washing, filtration, and centrifugation, ionic strength, temperature, and stirring speed. Different protein/polysaccharides and lipopolysaccharide combinations have been employed for the encapsulation of essential oils using the method of complex coacervation, such as gelatin/gum arabic, gelatin/sodium alginate, gelatin/pectin, sodium alginate/chitosan, milk protein/carboxymethyl cellulose, whey protein isolate/gum arabic, whey protein isolate/sodium carboxymethyl cellulose, whey protein isolate/sodium alginate, chitosan/kappa carrageenan, chitosan/gum arabic, chitosan/sodium carboxymethyl cellulose [31,32,33,34,35,36,37,38,39,40].

Nanofibers are obtained by a process that implies the conversion of biopolymer solution into nanofibers using electrohydrodynamic techniques such as electrospraying (low-density polymer solution) and electrospinning (high-density polymer solution). The fibers are formed under an electrical field passing through the syringe. The first step in the preparation of nanofiber-loaded essential oils implies the formation of EO emulsion in a hydrophilic biopolymer solution using usually non-ionic surfactants from the group of Triton X or Tween. Active packaging based on nanofibers obtained from chitosan, gelatine, alginate, zein, chitosan/gellan, tragacanth, and sodium alginate/polyvinyl alcohol with different EOs has been proven effective in the protection of food products against foodborne pathogens [41,42,43,44,45,46].

**Figure 1 antibiotics-12-01473-f001:**
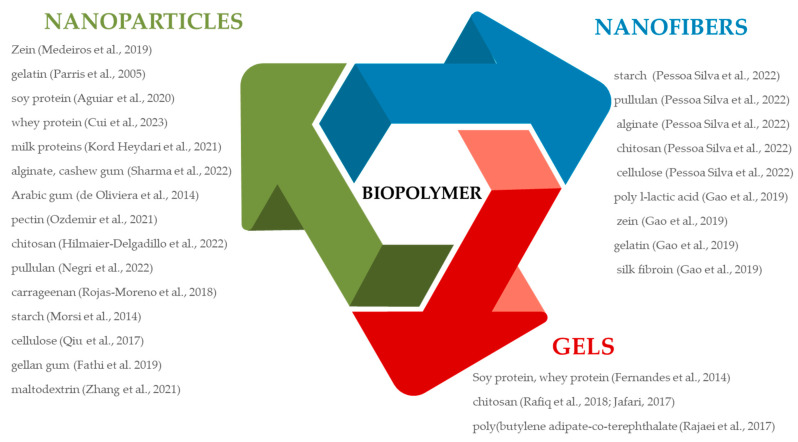
Biopolymer-based structures for encapsulation of essential oils [37,47,48,49,50,51,52,53,54,55,56,57,58,59,60,61,62,63,64,65,66].

3D networks are comprised of chemically or physically crosslinked hydrophilic or amphiphilic polymers, usually nano-sized (1–200 nm), called nanogels. According to affinity for certain solvents (water or hydrophobic ones), nanogels are divided into hydrogels and organogels. Hydrogels are obtained by the crosslinking of hydrophilic monomers or polymers, possess an affinity for water, and are suitable for the incorporation of hydrophilic compounds. On the other hand, organogels possess an affinity for a hydrophobic medium and are suitable for entrapping essential oils. Nanoorganogels have been proven to be promising systems for encapsulating and delivering EOs, improving their performance. They are obtained by a covalently crosslinked or self-assembly process. Nanogels based on chitosan/myristic acid and chitosan/benzoic acid have been shown to be efficient in the encapsulation of clove and rosemary essential oils, contributing to the better antimicrobial activity of active packaging toward food pathogens [47,48]. The high encapsulation efficiency of oregano essential oil was achieved using a procedure of ionic gelation to entrap EO into the alginate-based hydrogels. This kind of hydrogel is obtained using spray-drying and subsequent ionic gelation by dripping microcapsules in a calcium chloride solution. Poly(butylene adipate-co-terephthalate) (PBAT)/cassava starch films containing obtained microcapsules were produced by extrusion blow molding, which has demonstrated the high efficiency of obtained gels to protect encapsulated EO from thermal degradation [49]. Availability, biocompatibility, and the ability to encapsulate antimicrobial compounds and improve the applicative properties of biopolymer-based films make clay one of the most preferred nanomaterials. They consist of aluminosilicate layers with traces of organic matter, metals, and metal oxide, parallel-bonded due to secondary forces. Clay minerals are classified according to their chemical composition into three main categories: the first one with a silica/alumina ratio of 2:1 (montmorillonite, bentonite, etc.), the second with a silica/alumina/magnesium ratio of 2:1:1, and the third with silica/alumina ratio of 1:1 (kaolin, halloysite) [67]. The entrapping of EOs in montmorillonite and halloysite is carried out via the evaporation/adsorption procedure by directly mixing of EOs with clay minerals, avoiding high temperatures. The encapsulation of carvacrol, oregano, and cinnamon essential oils into halloysite-based nanocarriers incorporated into cardboard boxes has resulted in improved antimicrobial activity against foodborne pathogens. Active packaging has been successfully manufactured by the incorporation of montmorillonite-loaded EOs into chitosan, poly(hydroxybutyrate-co-hydroxy valerate), hydroxypropyl methylcellulose, and alginate; bentonite–loaded EOs into chitosan, cassava starch, levan, and poly(ε-caprolactone); and kaolinite into poly(hydroxybutyrate-co-hydroxy valerate) [68].

Another example of a good carrier for antimicrobial substances are the cyclodextrins (CDs). CDs are cyclic oligosaccharides obtained as a result of the enzymatic conversion of starch by the action of the enzyme glycosyltransferase, produced by the bacteria *Bacillus* sp. and *Klebsiella*. The main types of cyclodextrins consist of 6, 7, and 8 glucopyranose units (α, β, γ) connected by a (1–4) glycosidic bond. The inner cavity of CDs is hydrophobic, while the outer side has an affinity toward the aqueous phase. With hydrophobic substances, such as essential oils, CDs form host-guest inclusion complexes via secondary forces, enabling their compatibilization with a hydrophilic biopolymer matrix. The procedure of preparation of inclusion complexes implies the dispersion of CDs and essential oil in an aqueous medium, stirring in mild conditions, and the filtration of obtained inclusion complexes [30]. Active packaging containing CDs can be prepared by solution casting (clove oil with chitosan/β-cyclodextrin citrate/oxidized nanocellulose biocomposite) [69], formation of edible coatings (chitosan-gelatine edible coating with β-cyclodextrin/lemongrass essential oil inclusion complex) [70], or CD-EO inclusion complexes can be directly subjected to the electrospinning technique to form a nanofibrous web with potential application in active packaging [71].

## 4. Antimicrobial Packaging as Control of Microbiological Activity in Food

The ideal antimicrobial biopolymer packaging should meet several important criteria. To begin with, all phases (components) in their production must, following legal regulations, meet the conditions of direct contact with food [72]. In addition, the procedure for incorporating the antimicrobial phase should be simple and economical, so that the final price of the active packaging would be attractive to the food industry [23]. Like any other packaging, this one must remain chemically stable for long-term use and storage and satisfy the function of a water and gas barrier. Also, packaging with antimicrobial action must not be subject to easy and rapid degradation (especially in the case of biopolymers) and should maintain its antimicrobial function during the entire packaging and storage process. No less important is the fact that this kind of packaging must not have a harmful effect on people’s health when handling it or consuming the food that was packed in it. The antimicrobial component added to this type of active packaging must not exceed the maximum allowed amount of antimicrobial agent and be in accordance with the type of packaged food [73].

The basic division of packaging with antimicrobial action is biodegradable and non-degradable. Most synthetic polymers are not biodegradable, which is why they are often used as packaging materials. In addition, they are characterized by low cost, low density, internality, good protective properties, mechanical strength, a high degree of transparency, the ability to heat seal, and the possibility of labeling and printing [74]. The most commonly used synthetic packaging materials in the food industry are polyethylene, polypropylene, ethylene vinyl acetate, polystyrene, polyethylene terephthalate, and polyvinyl chloride. Synthetic biodegradable polymers such as polycaprolactone or polyvinyl alcohol are also used, which are still in the minority compared to non-degradable ones [73,75]. Additionally, this would mean that the use of packaging based on synthetic polymers has a negative impact on the environment via the creation of landfills, environmental pollution, and high levels of energy consumption during their production. For these reasons, the use of biopolymers and materials of natural origin is being sought, which are currently a more expensive but more environmentally friendly solution due to their biodegradability [76]. For example, the use of biopolymers from insufficiently used materials or agro-industrial and food by-products and waste is increasingly recommended, which would significantly reduce the costs of biopolymer packaging [77].

## 5. Incorporation of Essential Oil as the Antimicrobial Substance in the Biopolymer Matrix

Essential oils are hydrophobic liquids consisting of volatile aromatic compounds with diverse effects—antimicrobial, antioxidant, anti-inflammatory, analgetic, anti-depressive, relaxing, etc. They are highly recognized in the pharmaceutical industry and medicine, but also in the gastronomy and food packaging industries. EOs are obtained by extraction from different plant parts—seeds, stems, bark, buds, leaves, and flowers. The composition and yield are influenced by the conditions of plant growth and harvesting as well as extraction parameters (type, solvents, duration, etc.). Among different extraction methods for EO, steam distillation is the most widely used method, resulting in a high yield (above 93%) [78]. This method along, with Eos, gives a secondary product—hydrosol or hydrolat, which contains water-soluble compounds [79].

It can be summarized that essential oils play a crucial role in active packaging systems by virtue of their profound antimicrobial attributes, which synergistically enhance the protective capabilities of packaging materials against microbial threats. These oils, harnessed from botanical sources, contain a rich array of bioactive constituents, including phenols, terpenes, and aldehydes, that collectively exhibit potent inhibitory effects against a spectrum of microorganisms. The main compounds of essential oils belong to the class of monoterpenes, monoterpenoids, and phenylpropanoids (with alcohol or aldehyde functional groups) [80]. Those compounds have antimicrobial effects. EOs are recognized as safe by the U.S Food and Drug Administration (FDA) [81], which, along with their antimicrobial effects against foodborne pathogens and use as food flavors, make them suitable for food packaging applications. To be used in active food packaging, essential oils should show activity against foodborne pathogens, which cause food spoilage, but also foodborne illness in people, which can have serious consequences. This effect has been proven for a wide spectrum of essential oils, such as eugenol essential oil, ginger essential oil (*Zingiber officinale*), cinnamon oil, thyme oil (*Thymus vulgaris*), lemongrass oil (*Cymbopogon citratus*), oregano oil, tea tree oil (*Melaleuca alternifolia*), sage oil (*Salvia officinalis*), catnip essential oil (*Nepeta cataria*), *Satureja* Khuzestanica, *Helichrysum italicum*, mustard (*Brassica nigra*), etc. [63]. Mechanistically, essential oils disrupt cellular membranes, impede critical enzymatic processes, and intricately modulate microbial gene expression. This intricate interplay of multifaceted mechanisms underscores their ability to hinder microbial proliferation, ensuring extended shelf life and enhanced product safety in the context of biopolymer-based active packaging systems [82]. This integration not only capitalizes on the antimicrobial potential of essential oils but also leverages the innate biocompatibility of biopolymers, thus engendering a harmonious synergy. The compatibility of essential oils with biopolymers ensures a controlled and sustained release of their antimicrobial constituents, effectively extending the protective capabilities of the packaging throughout the product’s shelf life. Moreover, the introduction of essential oils can mitigate the drawbacks associated with synthetic additives while addressing consumer preferences for natural and eco-friendly packaging solutions [83]. Different approaches have been developed for the incorporation of essential oils in a biopolymer matrix (direct incorporation, encapsulation), which depend on the compatibility of essential oils and biopolymers.

### 5.1. Direct Incorporation

When a biopolymer possesses a hydrophobic character, such as in the case of cellulose acetate [84], cellulose acetate/polycaprolactone diol [83,85], or polylactide [86], EOs can be directly and uniformly incorporated into the biopolymer matrix. Some researchers have carried out direct incorporation of essential oils into a hydrocolloid matrix (cassava starch/cellulose nanofiber, sago starch, starch/chitosan, chitosan/gelatine, neat chitosan, etc.) without the addition of amphiphilic molecules using a high-sheared homogenizer (over 10,000 rpm) [87,88,89,90]. However, this method can lead to the accumulation of essential oil on the surface of the film because of the differences in polarity between essential oil and hydrocolloids. That changes the kinetics of releasing active compounds to the food surface, decreases their activity, and can also lead to the EOs oxidation if they form a layer on the top surface of the film. However, some proteins (whey proteins) have an emulsifying effect, enabling the formation of a uniform dispersion of EOs in a hydrocolloid matrix [91].

In order to achieve complete incorporation (solubility) of essential oils into hydrocolloids—water-soluble proteins and/or polysaccharides matrix, controlled release of EOs, their protection against oxidation, thermal degradation, and evaporation—with the aim of ensuring uniform dispersion into a hydrocolloid matrix, it is necessary to encapsulate them using different approaches such as emulsification, liposome formation, entrapping in biopolymers and clay-based nanocarriers, as well as in cyclodextrins. The advantages and limitations of each method of incorporation of essential oils in a biopolymer matrix are given in Table 4.

### 5.2. Emulsification

Encapsulation by emulsification is a simple method for the compatibilization of essential oils and hydrophilic biopolymers, which requires high mechanical energy incorporated in the system and the addition of amphiphilic molecules—emulsifiers or surfactants. After the formation of EO droplets in the aqueous medium under high-speed stirring, an emulsifying agent intended for the formation of oil-in-water emulsion is added. The droplet size is mostly between 20 and 200 nm; therefore, the obtained emulsion has the prefix nano [92]. Different researchers have used surfactants (oil in water)—synthetic, mostly non-ionic Tween 20 and Tween 80, and natural ones (soya lecithin). This procedure has been applied by researchers who developed active packaging by incorporating different essential oils into films based on gelatin [93], fish gelatin [94], gelatin-chitosan [95], chitosan [96], chitosan–carboxymethyl cellulose [97], carboxymethylcellulose [98], sodium alginate/carboxymethyl cellulose [99], methylcellulose [100], whey protein isolate [101], whey protein isolate incorporated with chitosan nanofibers [102], etc. This method consists of several steps, where the first implies the dissolution of surfactant into water and the addition of essential oils to form an emulsion, followed by the addition of the obtained emulsion/nanoemulsion to the water solution of biopolymers. The formed dispersion can then be subjected to the formation of nanofibers or films by the casting method, or, in the case of edible films, directly applied to the food surface. Also, the emulsification of essential oils into a biopolymer matrix can be performed by the subsequent addition of EO and emulsifier to the biopolymer solutions and the application of ultrasound after intensive homogenization [103]. Instead of surfactants, Feng and co-authors have used starch modified by octenyl succinate anhydride with amphiphilic character to make nanoemulsion, which was incorporated into a pullulan solution to obtain biodegradable films [104].

### 5.3. Liposomes Formation

Liposomes are vesicles that consist of a hydrophilic core and an amphiphilic—phospholipids-based bilayer. Liposomes are categorized according to their size and lamellarity, such as small unilamellar vesicles (SUV)—with sizes in the range between 20 and 100 nm; large unilamellar vesicles (LUV)—with sizes above 100 nm; and multilamellar vesicles (MLV)—sizes above 500 nm [105]. Hydrophobic compounds, such as essential oils have been incorporated into the lipid bilayer. A procedure for the encapsulation of thyme essential oil in a liposomal chitosan-based matrix has been carried out by Al-Moghazy et al. Multilamellar and unilamellar vesicles were prepared by dissolving lecithin and essential oil in the chloroform, forming dry film, and adding chitosan solution to form dispersion, which was then subjected to stirring and sonication [106].

## 6. Migration of the Antimicrobial Substance into the Packaging System

The basic property of most antimicrobial packaging is based on the migration of the antimicrobial substance from the packaging matrix to the packaged food and/or the space around the food. In contrast to microbiological contamination, the migration of the active compound from the matrix is a deliberate, targeted, and necessary process to achieve the antimicrobial effect and protection of the antimicrobial package. Only proper and controlled migration of the antimicrobial substance can inhibit or reduce the proliferation of microorganisms and delay food spoilage and expiration [107].

Unlike other types of active packaging materials where the migration of the target compound does not have to be efficient, this property of antimicrobial packaging is a critical point and is largely determined by the proper release of the antimicrobial compound from the active material. Only a gradual migration of the antimicrobial agent will maintain an effective antimicrobial concentration of the target substance in the packaging system during a longer period of storage [108]. Behind this concept is the fact that antimicrobial agents must reach each microbial cell and act on them, while other compounds, such as antioxidants, can be effective without direct contact with the product and without a gradual release from the packaging matrix. A properly designed antimicrobial packaging system can protect packaged food during the long-term process of transportation and storage, with a minimal concentration of the antimicrobial agent in the food product itself [107].

However, there is one important but expected drawback of antimicrobial packaging. In addition to the desired migration of the active compound, antimicrobial packaging can also release other low-molecular-weight substances into the food. Volatile substances, polymer additives, residual monomers, or oligomers that are not bound to the packaging matrix can migrate into the packaged food [109]. The migration of undesirable molecules can change the sensory acceptability of food or even pose a risk to the consumer. Therefore, European Regulation 10/2011/EU established a regulation that packaging material that poses any risk to the health of consumers must not be included in circulation. This regulation establishes global and specific limits for concentrations of substances that migrate from packaging to packaged food. In relation to active materials, antimicrobial packaging belongs to the packaging systems covered by this regulation [110].

The term “migration” refers to the transfer of low-molecular-weight compounds from the packaging matrix to the packaged food. In polymer packaging, the migration process is characterized by great complexity, which is strongly influenced by the interaction between the food components and the packaging material [109]. Bhunia et al. [111] explained the migration mechanisms, dividing them into four main steps:Diffusion of the substance from the packaging matrix;Desorption from the matrix surface;Sorption of the substance in the contact space;Desorption into food.

This migration mechanism can be twofold, depending on whether the packaged food is in direct contact with the packaging matrix or not (Figure 2). Namely, when a packaging system is established that contains an intermediate space between the polymer matrix and the food, we speak in indirect contact, and the migration mechanism is changed. In this case, there are additional steps in the migration mechanism that involve the diffusion of migrants into the interstitial space and mass transfer through the second contact surface to the food [112]. The phenomenon of migration includes two thermodynamic and kinetic parameters, namely the distribution coefficient and the diffusion coefficient [113]. The diffusion coefficient is determined by mass transfer due to the movement of molecules from an area of high concentration to an area of low concentration until equilibrium is established. The distribution coefficient refers to the concentration balance of the substance that migrates between the packaging and the packaged food. Both coefficients will depend on the properties of the material, the antimicrobial substance, and the packaged food. All substances present in the packaging system can migrate independently of the layer and the position they occupy. Therefore, the method of production of the packaging (polymer, properties, polarity, etc.), the characteristics of the antimicrobial substance (washability and polarity), the chemical and temporal interaction between the antimicrobial substance and the polymer matrix, the characteristics of the food (composition, pH value, and humidity), and environmental factors (temperature and relative humidity) must be examined before commercial use [113].

For antimicrobial packaging to be effective, the active compound must be released at a minimum inhibitory concentration (MIC) level. However, this reference concentration will not only depend on the antimicrobial activity of the compound incorporated in the polymer matrix but also on the matrix itself, given that the interaction between them strongly affects the release of the active substance [114]. Therefore, the method of incorporating the active compound into the polymer matrix is extremely important. Also, when it comes to the mechanisms of migration and the interaction of the antimicrobial substance and the polymer matrix, the nature of the polymer used (natural or synthetic) plays an important role [114]. The effect of temperature and relative humidity on migration arrest is much greater in the case of biopolymer-based packaging systems compared to synthetic polymers due to the hydrophilic nature of some protein- and carbohydrate-based materials [115].

It is important to emphasize that multi-layer systems provide numerous advantages in terms of the migration of the active substance in the packaging. As for the migration process, the combination of different layers achieves a controlled release of the active compound from the packaging. From an industrial point of view, this combination can affect packaging costs, which can be another disadvantage of active packaging [114].

## 7. The Influence of the Packaging System on the Inactivation of Microbiological Contamination

In the last decades, materials for food packaging have been improved in several different aspects, such as the development and selection of the best packaging option, prolongation of the shelf life of food, meeting environmental requirements, etc. Modified atmosphere-, smart-, and active packaging have become essential in the improvement of food storage and protection conditions from processing and production processes, through handling and storage, to the final consumer and disposal [116]. Interestingly, about 25% of the total costs of the consumer food industry are packaging costs. It is the reason the incentive for the production of functional packaging with minimal costs and minimal impact on the environment is a priority for this century. Disciplines such as chemistry, microbiology, food science, and engineering must be involved in the production of any type of modern packaging to realize the prerequisite of an interdisciplinary base in the field of packaging materials technology [18].

Furthermore, packaging materials should not be seen as waste but as a resource. One of the best ways to achieve this result is by recycling. Many food companies have been much more interested in another way of using packaging materials: biodegradable packaging. However, a major problem is excessive costs in proportion to the amount of biodegradable material that has to be used [27]. Equally valuable are biopolymer materials that are increasingly used for the partial or complete production of packaging materials that are inert to the food matrix [26]. The time of use of food packaging is very short, so it is not economically convenient to spend a lot of money on the production of innovative materials only for the preservation and transportation of food with a short shelf life. The use of biodegradable or biopolymer materials could be a good alternative to reduce environmental impact. The choice of the most suitable packaging depends on several factors, primarily on the type of food that must be protected. Some of the most important properties of packaging are mechanical, physical-chemical, and optical properties, which play a key role in choosing the appropriate material [26,27,116,117].

Today’s trends in developed countries related to fresh, minimally processed, easy-to-prepare, and RTE foods represent major challenges in terms of the microbiological safety and quality of these foods. A similar problem arises in the course of globalization, i.e., the distribution of food from centralized industrial systems to the whole world [15]. This flow of the food industry also means the production of food with an extended shelf life to ensure the global market. Consequently, the presence and potential of microbiological contamination are extremely high, reducing shelf life and increasing the risk of foodborne illness. Therefore, the entire food industry is directed towards innovative ways of inhibiting microbial growth in food while maintaining quality, freshness, safety, and expiration dates [118].

Although the primary idea of inactivating microbiological contamination was related to improvements in traditional food preservation methods (drying, heating, freezing, fermentation, salting, etc.), it was quickly realized that no single method is comprehensive and is not the right solution for inhibiting the growth of microorganisms and preventing food spoilage completely. One of the innovative strategies to extend the shelf life of food is the addition of antimicrobial substances to food, which has also had a limited effect given that most consumers want high-quality food without preservatives [119]. Also, the direct surface application of antimicrobial substances to food is restricted due to the rapid diffusion from the surface into the food mass, which causes only a momentary reduction of the microbial population [120]. However, this strategy was retained in the food industry but received a different performance. Namely, the research on antimicrobial agents of natural origin that prevent the growth of microorganisms is aimed at the use of special packaging systems with an antimicrobial effect to ensure increased food safety and quality [117].

As already mentioned, the primary function of packaging is related to the protection of food from external influences, mechanical force, and spoilage by environmental microorganisms, moisture, gases, dust, odors, etc. [117]. Also, the packaging is very important from the point of view of marketing and standardization because it can be used as an informative asset for consumers. Sung et al. [117] emphasize that in this way useful information is provided to consumers, and the product itself is made more usable and practical. On the other hand, the antimicrobial food packaging system is a modern concept of active packaging and represents one of the most perceptive concepts for food safety and quality [121].

Packaging with antimicrobial activity is reflected in a physical combination of two or more chemically different phases, where one of the phases is the matrix of the packaging, while the other phases are in the form of embedded substances with antimicrobial properties [122]. One important advantage of antimicrobial packaging is that it is a more effective system than directly adding an antimicrobial agent to food. The reason for this advantage is the time-limited and controlled release of the antimicrobial substance or the identical but direct contact with food, which is achieved by different methods of addition and retention of the phase in the basic matrix phase of the packaging [117].

In other words, the use of polymeric materials as a matrix phase (carrier) of antimicrobial agents has a consequent, threefold role, as presented in Figure 3 [120].

According to Radusin et al. [121], antimicrobial substances incorporated into packaging systems have one of three types of mechanisms of action on microorganism cells: destruction of the cell wall and/or membrane, inhibition of enzymes in the cell, or destruction of genetic material. Antimicrobial substances include inorganic metal compounds (silver, titanium dioxide, zinc oxide, magnesium oxide, etc.), organic acids (sorbic, citric, propanoic, etc.), enzymes (lysozyme, lactoferrin, etc.), bacteriocins (nisin, pediocin, etc.), fungicides (benomyl, imazalil, etc.), poymers (chitosan), natural extracts (bay leaf, oregano, lemon grass, etc.), antibiotics (natamycin), surfactants (lauric arginate), and other compounds (phenolic, Maillard reaction products, etc.). Regardless of which substance it is, the selection of suitable packaging material should be based on its spectrum and mode of action, chemical composition, and speed of action [122,123].

It most often happens that a certain antimicrobial substance is effective only on one group of microorganisms. That is why silver ions are often used due to the wide spectrum of antimicrobial activity against gram-negative and gram-positive bacteria, fungi, protozoa, and certain viruses [122,124]. In addition to silver, metal oxides such as TiO_2_, ZnO, and MgO have antibacterial activity attributed to their effect on the cell’s respiratory system (creating reactive oxygen). Titanium dioxide is non-toxic and has been approved by the FDA for use in food, drugs, and food contact materials [125]. On the other hand, ZnO particles in contact with microbial cells can act bacteriostatically or bacteriocidally depending on the concentration of the compound, but they act equally well on gram-positive and -negative bacteria [122]. For now, the only assumed bactericidal mechanism of MgO is related to the production of high concentrations of superoxide anions on its surface, which react with the carboxyl groups of peptide bonds in bacterial cells, thereby destroying them [122]. Antimicrobial activity has been observed with other metals, semimetals, and alkaline earth metals, especially in the form of nanoparticles (CdSe, CdTe, Au, Al_2_O_3_, and iron oxides) [117].

In recent years, the use of inorganic nanoparticles as antimicrobial substances in packaging materials has particularly intensified. This is attributed to the good stability of these materials and their ability to withstand process conditions such as high pressures and temperatures in plastic manufacturing processes [126]. In addition, organic acids have a long history as food preservatives with GRAS status, most of which do not have an acceptable daily intake limit for humans. As for enzymes as antimicrobial agents, lysozyme is the most widely used. It is often incorporated into packaging films, where, in contact with the microbial cell, it hydrolyzes or dissolves the cell wall [127]. Also, recent scientific works suggest that the effect of lysozyme is enhanced in the presence of detergents and chelates, or that the enzymatic effect of this substance can be enhanced by the addition of lactoferrin [117]. As a bacteriogenic product, bacteriocins have gained GRAS status and are used as natural antimicrobial substances. For example, nisin, which is produced by Lactococcus lactis, can protect this bacterium against Listeria, Staphylococcus, Bacillus, and Clostridium. Nisin is an interesting choice for incorporation into antimicrobial packaging on an industrial level, as it interacts with the microbial surface and does not need to be incorporated into the matrix to exert its effect [128]. Natamycin is a natural antifungal agent produced by the bacterium Streptomyces natelensis during fermentation. It is widely used in the food industry to prevent fungal contamination in meat, cheese, and fruit. It is approved as a food additive (E235) in over 40 countries, carrying GRAS status. Natamycin is an effective compound that acts specifically by binding to ergosterol, thereby inhibiting vacuolar fusion. Due to this mechanism of action, it is active against molds and yeasts but not against bacteria and viruses [129]. Natamycin has an advantage over other preservatives because it does not affect the taste and the appearance of the final product [130].

A wide variety of natural extracts, such as pigments, garlic, lemongrass, cinnamon, cloves, and oregano, have shown antimicrobial activity in packaging material against a large number of bacterial pathogens [131]. The most common mechanism of action of these substances is disruption of the cell and mitochondrial membrane, which leads to disruption of the cell membrane, cytoplasmic leakage, cell lysis, and, finally, death. However, the negative side of adding plant extracts is the possibility of adding them in low concentrations due to their effect on the smell and taste of food. In addition, these extracts are susceptible to lipid oxidation [132].

Finally, it should be emphasized that the direct addition of an antimicrobial substance to food can reduce food quality, changing the organoleptic and textural qualities of food [117]. Consequently, packaging with an antimicrobial effect will play a very important role in inhibiting the growth of targeted microorganisms on/in food while simultaneously improving food safety and extending shelf life without loss of quality.

## 8. Proposal for Comprehensive Determination of Antimicrobial Potential of Biopolymer-Based Active Packaging with Incorporated Essential Oils

While in vitro studies provide valuable insights into the antimicrobial properties of biopolymer-based active packaging, they have limitations and should be complemented with in vivo experiments on the specific food products the packaging is designed for. Generally, when it comes to testing the antimicrobial potential of newly designed biopolymer-based active packaging with incorporated essential oils, researchers should follow these steps:Preliminary antimicrobial analysis of the selected essential oils by disc-diffusion method;Determination of the minimal inhibition concentration (MIC) of the selected essential oils;In vitro assessment of the antimicrobial activity of active packaging systems;In vivo assessment of the antimicrobial activity of active packaging systems;Sensory analysis.

As presented in Figure 4, the first mandatory step in testing the antimicrobial potential of active packaging prepared with various essential oils should be the examination of the antimicrobial activity of the essential oils themselves. The most common approach when testing the antimicrobial activity of the essential oils is to first examine them using the disc-diffusion method. This method is relatively cheap and easy to perform, so it is a choice of many authors [133,134,135,136]. The method consists of several steps: (i) activation of the chosen microorganisms on a suitable culture medium; (ii) preparation of cell suspension with an approximate concentration of 10^6^ CFU/mL; (iii) homogenization of 1 mL of cell suspension with 9 mL of melted and tempered culture medium, which should be poured into Petri dishes; (iv) after the solidification, the sterile discs (6 mm in diameter) should be placed onto the inoculated medium; (v) on each disc, an aliquot of 15 μL of the tested essential oil and suitable positive and negative controls is applied; and (vi) after the incubation period, the diameter of the halo zone can be measured for each disc and expressed in mm [134].

In order to obtain the minimal inhibitory concentration (MIC) of the selected essential oil capable of inhibiting microbial growth up to 90%, the microdilution method is usually employed. The preparation of cell suspension is performed in the same manner as the previously described disc diffusion method. Afterwards, the serial dilutions of the tested essential oil were prepared, and equal volumes (100 μL) of each dilution and inoculated media were transferred to a sterile flat-bottom, 96-well microtiter plates. After the period of incubation, 100 μL of mixture from each well was poured into Petri dishes and homogenized with suitable culture media. Petri dishes are commonly incubated under the same conditions as microtiter plates, and the grown colonies are enumerated by viable count. The final results of the MIC determination are calculated as follows:$$100\times \frac{\left(\;N_{c}-N_{t}\right)}{N_{c}}\;\;(\%)$$
where *N_c_* is a number of cells in a positive control and *N_t_* is a number of cells after the contact of essential oil dilution and microbial cells. This method seems to be indispensable when testing compounds with promising antimicrobial effects [137,138,139].

In vitro assesment of antimicrobial activity of active packaging systems involves conducting laboratory experiments to evaluate the effectiveness of each packaging layer in inhibiting the growth of microorganisms that can spoil or contaminate food products. In this step, it is very important to test each layer individually and their combinations to ensure their activity against the selected microorganisms [70]. Also, the choice of relevant microorganisms that are commonly associated with food spoilage or contamination is crucial. When the packaging materials and microorganisms are chosen, the disc diffusion method and the determination of minimal inhibitory concentration should be performed as previously described. It should be pointed out that control experiments must always be an integral part of the experiments related to the evaluation of antimicrobial activity. When testing active packaging systems, control experiments refer to the use of packaging materials without antimicrobial agents to establish a baseline for microbial growth inhibition. Also, the testing is completed using positive controls by using known antimicrobial substances (antibiotics, antimycotics, etc.) to validate the experimental setup. Finally, it is important to perform multiple replicates of the experiments to ensure the reliability and statistical significance of the results. Some authors suggest that when preparing the active packaging systems with two or more antimicrobial agents, their synergistic effects have to be examined [140,141]. By mixing various antimicrobial agents and through their synergistic activity, many advantages can be expected: (i) induction of stronger antimicrobial activity; (ii) the extension of the spectrum of antimicrobial action; and (iii) the prevention and suppression of the regrowth of undesirable microorganisms [141]. However, further analysis, including in vivo trials, are essential to validate the exact effectiveness of active packaging systems under practical conditions.

In vivo assessment of the antimicrobial activity of active packaging systems involves conducting experiments in real-life conditions to evaluate the effectiveness of packaging materials in inhibiting the growth of microorganisms and preserving the quality and safety of food products. Unlike in vitro assessments, which are conducted in controlled laboratory settings, in vivo assessments provide a more realistic understanding of how active packaging systems perform in practical situations. After choosing the relevant microorganisms or pathogens that are likely to affect the quality and safety of the specific food being packaged, their suspension should be made, and the selected food should be artificially contaminated with them. Afterwards, the active packaging materials with antimicrobial agents can be placed on the selected food, ensuring direct contact with it. The packaged food is then stored under the conditions that simulate the intended storage environment (temperature, humidity, etc.), and periodically, the samples of the packaged food are performed to assess microbial populations over time. The common microbial analyses that are conducted for this purpose are colony counting by using standardized methods [142], PCR (polymerase chain reaction), or some statistical analysis to avoid more complex destructive methods of analysis [70]. Also, in this type of analysis, comparative studies should be incorporated. Comparative studies are related to testing the same food product using traditional or non-active packaging systems at the same time and under the same conditions as the active packaging systems. This allows a direct comparison of the antimicrobial effects of the newly designed active packaging system with the old-fashioned one. Since essential oils, hydrolates, or some plant extracts are commonly used as active ingredients in active packaging, sensory analysis must be conducted when such a packaging system comes into direct contact with food.

Sensory analysis of active packaging systems involves evaluating how the packaging materials affect the sensory attributes (taste, smell, appearance, and texture) of the packaged food products. This type of analysis helps determine whether the active packaging materials introduce any changes in the sensory qualities of the food, which can influence consumer perception and acceptance. In order to perform sensory tests, a panel of trained sensory analysts or consumers who are skilled in evaluating food products should be recruited [143]. The sensory analysts should consume food products prepared with and without an active packaging system. It is very important to ensure that the packaging is consistent and that any differences are solely due to the active packaging system. Also, to prevent bias, the samples should be coded or numbered and randomized during the presentation to the sensory panel. For this purpose, different sensory tests can be used [144]:Difference Test—determine if there are detectable differences between food samples packaged with and without active ingredients;Descriptive Analysis—evaluate specific sensory attributes (e.g., taste, odor, texture) and quantify the intensity of these attributes using a trained panel;Consumer Acceptance Test—Assess consumer preferences and overall acceptance of the food products with active packaging systems compared to controls.

The results obtained by the sensory panel should be interpreted and analyzed to determine if there are any noticeable sensory changes introduced by the active packaging system, taking into account whether these changes are acceptable to consumers or if they may impact the marketability of the product. Moreover, the results of sensory analysis can be compared with the data obtained during microbial analysis in order to understand if any changes in sensory attributes correspond to changes in microbial populations.

Sensory analysis of active packaging systems is crucial for ensuring that the packaging materials do not adversely affect the sensory characteristics of the food products. This information is essential for product development, quality control, and consumer acceptance.

## 9. Conclusions and Future Perspectives for Real Application

The incorporation of essential oils as antimicrobial substances within biopolymer-based active packaging has demonstrated the packaging material’s antimicrobial efficacy. The bioactive compounds inherent in essential oils exhibit pronounced inhibitory effects against a spectrum of pathogenic and spoilage microorganisms, thereby manifesting a tangible extension of the shelf life of packaged commodities. This innovative approach offers a more sustainable alternative by mitigating reliance on conventional synthetic antimicrobial additives, thereby aligning with the contemporary drive for sustainable packaging modalities. The strategic incorporation of essential oils into biopolymeric matrices constitutes a propitious intervention, effectively attenuating the reliance on chemical preservatives and satisfying the increasing demand for environmentally based packaging solutions. The consequential elongation of product shelf life is a noteworthy outcome of this integration, notably benefiting perishable commodities encompassing many food items. The noticeable inhibition effect on spoilage microorganism proliferation is crucial in solving the perishability predicament that compromises product quality. Moreover, the incorporation of essential oils can offer ancillary advantages, particularly in the realm of flavor preservation. The aromatic origin of essential oils imparts a natural flavor profile to packaged foods, thereby augmenting their sensorial allure.

By addressing these critical fronts, the integration of essential oils within biopolymer-based active packaging is poised to substantiate its role as a pivotal stride towards efficacious and sustainable food preservation patterns. However, the journey is not without challenges, especially concerning essential oil volatility and compatibility within diverse biopolymer matrices. The vulnerability of essential oils to gradual volatilization underscores the need for meticulous considerations to ensure sustained antimicrobial efficacy. Furthermore, the homogenous dispersion or encapsulation of essential oils within packaging materials necessitates precise techniques due to the intricate nature of the process. The regulatory landscape also requires attention, as the utilization of essential oils in active packaging calls for adherence to rigorous safety and food contact standards. The variability in permissible concentrations across jurisdictions accentuates the need for strict regulatory compliance. The exploration of synergistic interactions among essential oils and other antimicrobial agents offers a promising avenue to enhance antimicrobial potency further. Understanding the mechanistic nuances underlying these interactions opens novel avenues to fortify the antimicrobial efficacy of active packaging materials. Moving forward, research endeavors should focus on optimizing formulation techniques, refining processing methodologies, and elucidating the long-term stability and controlled release kinetics of essential oil-infused packaging materials under varying storage conditions. Addressing these critical aspects will underpin the integration of essential oils within biopolymer-based active packaging as a pivotal step toward effective and sustainable food preservation strategies. The symbiotic alliance between essential oils and biopolymers holds immense potential for redefining packaging’s role in ensuring food safety and longevity while aligning with evolving environmental paradigms. Sensory evaluations discerning alterations in sensory attributes and preferences are imperative in refining the formulation of biopolymer-based packaging fortified with essential oils. Prospective research trajectories should endeavor to optimize formulation methodologies, augment processing techniques, and illuminate the long-term stability and controlled release kinetics of essential oil-infused packaging across varying storage conditions.

In conclusion, the integration of essential oils as potent antimicrobial agents within biopolymer-based active packaging materials represents a transformative advancement in the realm of food preservation and packaging. The incorporation of essential oils and biopolymers synergistically marries the innate antimicrobial attributes of the former with the structural advantages and eco-friendliness of the latter. This convergence not only bolsters packaging’s role in safeguarding product quality and shelf life but also resonates with the global call for sustainable packaging solutions. Crucially, the selection of appropriate biopolymers is an essential determinant of the success of active packaging systems. Different biopolymers bring distinct mechanical, barrier, and compatibility properties to the table, influencing the overall performance and practicality of the packaging. The adaptability of essential oils to diverse biopolymers is a testament to their versatility, with each combination offering a unique and tailored approach to combating microbial proliferation. Furthermore, the dynamic landscape of consumer preferences underscores the need for a holistic evaluation of the sensory attributes imparted by essential oils in packaged products. As consumers increasingly seek natural and minimally processed alternatives, the interplay between essential oils, flavors, and product quality gains paramount significance. Looking ahead, continued research is indispensable in fine-tuning the formulation and processing techniques of essential oil-infused packaging materials. The optimization of factors like essential oil concentrations, release kinetics, and shelf stability will pave the way for robust and scalable applications. Collaboration across multidisciplinary domains, encompassing chemistry, packaging engineering, microbiology, and consumer studies, holds the potential to yield comprehensive solutions that address both technical and consumer-driven challenges. The alignment of industry efforts and academic exploration will drive innovation in antimicrobial active packaging, propelling it from a nascent concept to a standard practice in ensuring food safety and extending shelf life. In summary, the integration of essential oils within biopolymer-based active packaging encapsulates a dynamic convergence of science, sustainability, and sensory appeal. By harnessing the power of nature’s antimicrobial arsenal and the ingenuity of biopolymers, this interdisciplinary approach forges a path toward safer, longer-lasting, and more environmentally conscious food products. The journey is ongoing, but the strides made so far signify a promising future where essential oil-infused packaging stands as a cornerstone in the modern food packaging landscape.

## Figures and Tables

**Figure 2 antibiotics-12-01473-f002:**
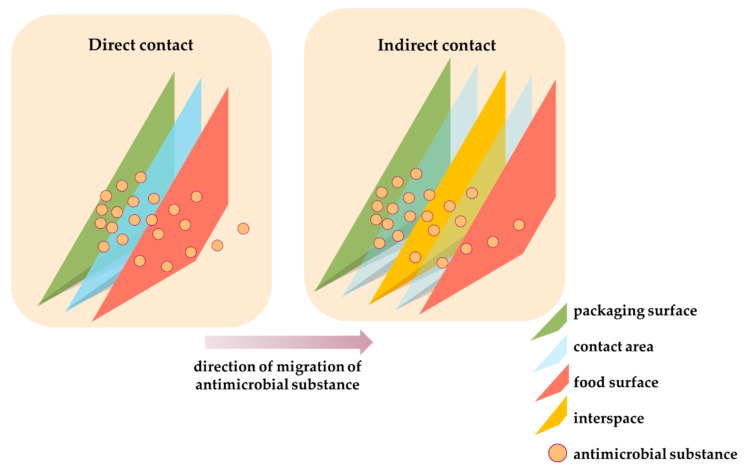
Direct and indirect contact between antimicrobial packaging and food sample.

**Figure 3 antibiotics-12-01473-f003:**
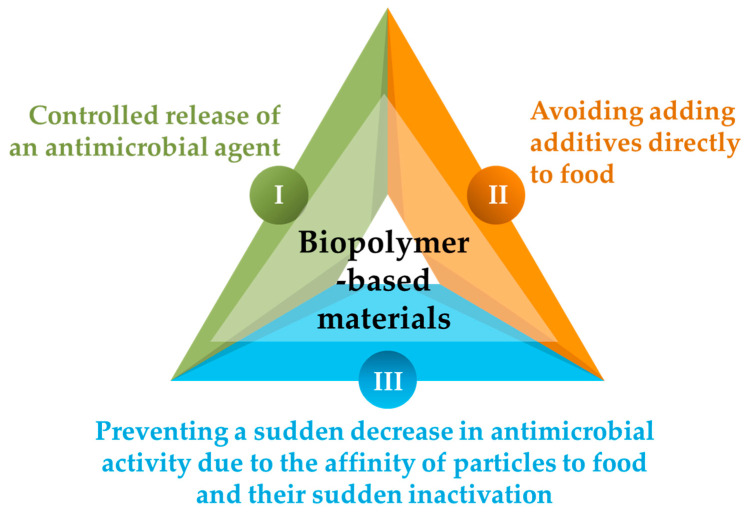
The requirements of biopolymer-based materials as a carrier for antimicrobials.

**Figure 4 antibiotics-12-01473-f004:**
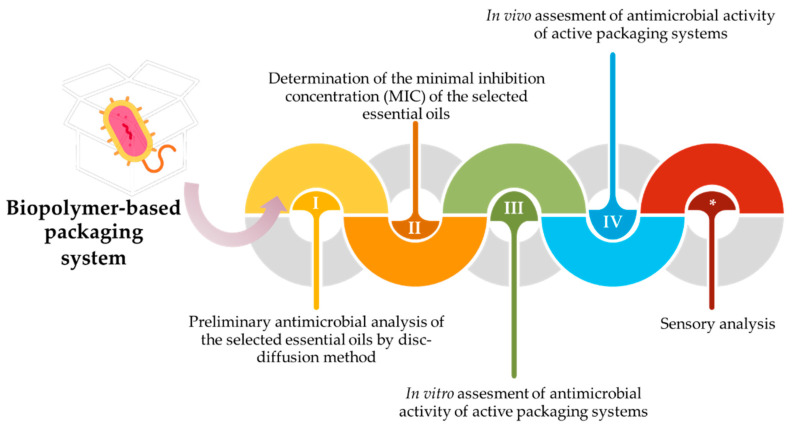
Traceability of steps in antimicrobial potential and sensory testing of packaging systems that involves biopolimer-based carrier and essential oil(s).

**Table 1 antibiotics-12-01473-t001:** Common foodborne bacteria.

Bacteria	Food Sources	Health Risks
*Salmonella*	Raw poultry, eggs	Gastroenteritis, fever
*Escherichia coli*	Undercooked beef, raw vegetables	Diarrhea, kidney failure
*Listeria monocytogenes*	Deli meats, soft cheeses	Listeriosis (severe illness)
*Campylobacter*	Raw or undercooked poultry	Diarrhea, cramps
*Clostridium botulinum*	Canned and low-acid foods	Botulism (paralysis)

**Table 2 antibiotics-12-01473-t002:** Common foodborne yeasts and fungi.

Microorganism	Food Sources	Health Risks
Yeasts		
*Saccharomyces cerevisiae*	Bread, beer, wine	Generally non-pathogenic
*Candida* spp.	Found in milk products	Opportunistic infections
*Zygosaccharomyces* spp.	Found in fermented foods	Generally non-pathogenic
*Debaryomyces* spp.	Dairy products, fermented foods	Generally non-pathogenic
Fungi		
*Aspergillus* spp.	Nuts, grains, dried fruits	Aflatoxin production
*Penicillium* spp.	Cheese, spoilage of various foods	Some species used in cheese production, allergenic spores
*Fusarium* spp.	Grains, cereals	Mycotoxin production
*Botrytis cinerea*	Fruits, vegetables	Allergenic spores
*Rhizopus* spp.	Fruits, vegetables, baked goods	Allergenic spores

**Table 3 antibiotics-12-01473-t003:** Packaging types based on used material(s), advantages, and limitations.

	Comparation of Packaging Type		
	Classical Food Packaging	Biodegradable Food Packaging	
Used materials	Metal Glass Paper Conventional plastic	Based on synthetic polymers	Based on biopolymers obtained from plants, animals, and microorganisms
Properties	Non-biodegradable persists in environment—degrades very slow good barrier and mechanical properties low levels of interaction with packaged food	Biodegradable in nature possesses worse mechanical and barrier properties low weight and ability to be easily handled	
Advantages	Strong and durable, excellent food protection longer shelf life of food product widely available and versatile; some of them can be low weight (plastic, paper) low levels of interaction with packaged food	Environmentally friendly, reduces plastic pollution can be produced from renewable resources nontoxic reduces carbon emissions and amount of plastic waste low weight and ability to be easily handled; can be edible can be designed to prolong food product shelf-life [29]	
Limitations	Non-environmentally friendly; production leads to great energy consumption, fossil fuel depletion, and emissions of volatile organic compounds (VOC) limited recyclability for some materials; long-time of decomposition; increase the amount of solid waste generation production of microplastic (conventional plastic)	Poor barrier and mechanical properties; may require specific disposal conditions to biodegrade properly (polylactide, polyhydroxyalkanoate)	

**Table 4 antibiotics-12-01473-t004:** The advantages and limitations of each method of incorporation of essential oils in a biopolymer matrix.

Method	Direct Incorporation	Emulsification	Liposomes Formation
The main characteristics of method	Using low energy if biopolymers possess hydrophobic character [63,64,65,66]; using a high-sheared homogenizer for hydrocolloids [67,68,69,70,71]	Using high-speed stirring and emulsifying agent intended for the formation of oil in water emulsion [73,74,75,76,77,78,79,80,81,82]	Dissolution of phospholipids and EOs in an organic solvent to create a lipid solution; evaporation of solvent and formation of a lipid dispersion in distilled water; size adjustment by ultrasonication.
Advantages	Shorter time for the preparation of active packaging; no additional steps –simpler procedure; can be more cost-effective	Stabilization and protection of EOs; improve their solubility in aqueous environments, provide their uniform distribution and controlled release. Nanoemulsions ensure better bioavailability of antimicrobial compounds and improved optical properties of packaging. In comparison to emulsions, nanoemulsions provide better stability to environmental conditions such as pH, temperature, and shear forces.	Targeted delivery and improved bioavailability, along with protection of ingredients and reduced undesirable reactions.
Limitations	Possible accumulation of essential oil on the surface of the film; deterioration of EO properties (oxidation, thermal degradation);	Specialized equipment and careful formulation to obtain stable formulation; Use of synthetic surfactants that can be irritable.	Complex production, difficulties in achieving consistent and desired liposome sizes, as well as the potential impact on texture, appearance, and flavor of food.
Method	Biopolymer–Based Nanostructures	Cyclodextrins	Nanoclays
The main characteristics of method	- Spray-drying by dispersing of EO in an aqueous solution of biopolymer obtain emulsion, which is pulled into the dryer through a nozzle giving small droplets (nanoparticles); - Complex coacervation when two oppositely charged biopolymers make an electrostatic complex (nanoparticles); - Electrospinning–formation of EO emulsion in aqueous biopolymer solution and fibers are formation of the fibers under an electrical field, passing through the syringe; - Covalently or self-assembly cross-linking of hydrophilic monomers or polymers (nanogels).	Mixing of EO and cyclodextrins in distilled water, stirring, filtration, and drying.	Preparation of EO and nano clay solution in an appropriate solvent. Mixing the solutions and stirring
Advantages	- Controlled release of active compounds, enhanced stability of essential oils, improved mechanical properties and durability (nanoparticles); - Can provide excellent barrier properties against gases and liquids; can be engineered to meet special requirements to gradually release antimicrobial compounds; can be transparent, which is important for packaging with special optical demands (nanofibers); - Can help maintain moisture levels in packaged foods, preventing dehydration and preserving freshness, and can improve barrier properties of packaging against oxygen and contaminants ensuring the controlled release of antimicrobial compounds over time (nanogels).	Odor and flavor control; help to maintain the sensory quality of the packaged food; they ensure protection of Eos’ degradation and their slow release over time.	Improvement of mechanical and barrier properties of packaging material.
Limitations	- Difficulties in achieving of uniform distribution, high cost of manufacturing, and incorporation into the packaging (nanoparticles); - Challenging scalability, commercialization, poor mechanical properties, and the high cost of their production (nanofibers); - Low level of compatibility between gels and matrix which may require some adaptations (nanogels).	High price and limited solubility in water.	Can cause reduced transparency and migration into packaged food.

## Data Availability

Not applicable.

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
