# Peer review of "Insight on Incorporation of Essential Oils as Antimicrobial Substances in Biopolymer-Based Active Packaging"

_antibiotics, 2023, doi:10.3390/antibiotics12091473_

Round 1
Reviewer 1 Report
The manuscript reviews all literature relevant data on incorporation of the essential oils in biopolymer matrix. The purpose of this incorporation is to exploit the antimicrobial properties of the essential oils for preserving food products' freshness and quality.
The introduction covers relevant literature data as well as the references are comprehensive. The manuscript is well organized describing among others different techniques for the incorporation of essential oils in biopolymer matrix and the emergence of microbiological contamination in the food industry.
The manuscript can be accepted for publication in its actual form.
Author Response
Reviewer 1
The manuscript reviews all literature relevant data on incorporation of the essential oils in biopolymer matrix. The purpose of this incorporation is to exploit the antimicrobial properties of the essential oils for preserving food products' freshness and quality.
ANSWER: The Authors would like to thank the Reviewer for a quick and professional review. The Authors believe that the actual paper would satisfy the scientific community and that it is going to be interesting enough for publishing in the Journal.
The introduction covers relevant literature data as well as the references are comprehensive. The manuscript is well organized describing among others different techniques for the incorporation of essential oils in biopolymer matrix and the emergence of microbiological contamination in the food industry.
ANSWER: Thank you for reviewing our paper and make effort to analyze all parts of our work.
The manuscript can be accepted for publication in its actual form.
ANSWER: The Authors would like to thank the Reviewer once again for proffesional review.
Reviewer 2 Report
The review article,"Insight on Incorporation of Essential Oils as Antimicrobial Substances in Biopolymer-based Active Packaging" by Ana Tomić et al, discusses the development of biopolymer-based active packaging with essential oil. The article was well organized and the flow was appreciable. I feel that this review article could be published in the journal without any further modifications.
Author Response
Reviewer 2
The review article,"Insight on Incorporation of Essential Oils as Antimicrobial Substances in Biopolymer-based Active Packaging" by Ana Tomić et al, discusses the development of biopolymer-based active packaging with essential oil. The article was well organized and the flow was appreciable. I feel that this review article could be published in the journal without any further modifications.
ANSWER: The Authors would like to thank the Reviewer for a quick and professional review. The Authors believe that the changed paper would satisfy the Reviewer' criteria and that it is going to be interesting enough for publishing in the Journal.
Reviewer 3 Report
In the antibiotics-2578706, the authors show detailed aspects regarding incorporating Essential Oils as antimicrobial substances in biopolymer-based active packaging. From 124 references, 57 have been published in the last 5 years. The abstract is well-structured, showing the fundamental problem and the proposed solution. The MS has 3 Figures well-realized and is organized as follows:
1. Introduction
2. Biopolimer-based structures
3. Incorporation of essential oil as an antimicrobial substance in the biopolymer matrix
3.1. Direct incorporation
3.2. Emulsification
3.3. Liposomes formation
4. Emergence of microbiological contamination in the food industry
4.1. Presence of bacteria in food
4.2. Presence of yeasts and fungi in food
4.3. The main problem related to the presence of microorganisms in food
5. The influence of the packaging system on the inactivation of microbiological contamination
6. Migration of the antimicrobial substance into the packaging system
7. Antimicrobial packaging as control of microbiological activity in food
8. Proposal for comprehensive determination of the antimicrobial potential of biopolymer-based active packaging with incorporated essential oils
9. Conclusions and future perspectives for real application
The following comments and suggestions are available below:
1. In the Introduction, the reviewer suggests following the succession from the abstract, developing microbial contamination as a significant concern (the current section 4) and the importance of essential oils as antimicrobial agents in the food industry. They are encouraged to show the main essential oils used for this purpose and their mechanisms of action.
2. They could continue evidencing the importance of the packaging system on the inactivation of microbiological contamination in a new Section 2, including the current sections 5, 6, and 7.
3. The types of packaging systems could be discussed in Section 3, thus justifying the biopolymer-based structure selection.
4. The authors are encouraged to summarize and compare the types of biopolymer-based structures and the methods of essential oils incorporation, briefly presenting the processes, advantages, and limitations.
5. The current Section 8 could become Section 4.
7. After all suggested changes, they are encouraged to reformulate suitable Conclusions and further perspectives.
Author Response
Reviewer 3
In the antibiotics-2578706, the authors show detailed aspects regarding incorporating Essential Oils as antimicrobial substances in biopolymer-based active packaging. From 124 references, 57 have been published in the last 5 years. The abstract is well-structured, showing the fundamental problem and the proposed solution. The MS has 3 Figures well-realized and is organized as follows: 1. Introduction; 2. Biopolimer-based structures; 3. Incorporation of essential oil as an antimicrobial substance in the biopolymer matrix; 3.1. Direct incorporation; 3.2. Emulsification; 3.3. Liposomes formation; 4. Emergence of microbiological contamination in the food industry; 4.1. Presence of bacteria in food; 4.2. Presence of yeasts and fungi in food; 4.3. The main problem related to the presence of microorganisms in food; 5. The influence of the packaging system on the inactivation of microbiological contamination; 6. Migration of the antimicrobial substance into the packaging system; 7. Antimicrobial packaging as control of microbiological activity in food; 8. Proposal for comprehensive determination of the antimicrobial potential of biopolymer-based active packaging with incorporated essential oils ; 9. Conclusions and future perspectives for real application
ANSWER: The Authors would like to thank the Reviewer for a quick and professional review as well as the opportunity to make essential and crucial changes in our work. All the Reviewer' remarks are accepted and the paper is changed according to their comments. The Authors believe that the changed paper would satisfy the Reviewer' criteria and that it is going to be interesting enough for publishing in the Journal.
We decided to revise the manuscript according to the Reviewer' remarks, highlighting the changes directly in the revised manuscript.
The following comments and suggestions are available below:
- In the Introduction, the reviewer suggests following the succession from the abstract, developing microbial contamination as a significant concern (the current section 4) and the importance of essential oils as antimicrobial agents in the food industry. They are encouraged to show the main essential oils used for this purpose and their mechanisms of action.
ANSWER: In accordance with the Reviewer's suggestion, the introduction of the manuscript are strategically changed to align with the progressive flow from the abstract while elucidating the pertinent concerns surrounding microbial contamination. We rotate sections, so we added previously section 4 after the Introduction part and make some changes. The part about essential oils are presented in currect section 4 which also involve the ways for their incorporations. Some crucial facts are added based on this Reviewer’s suggestion. Due to the fact that each chemical compound presence in essential oils can have different and several mechanisms of action during antimicrobial effect (many of them are not proof at all) we believe that this is not topic of this review, while the addition of that section will be disrupt the current workflow.
- They could continue evidencing the importance of the packaging system on the inactivation of microbiological contamination in a new Section 2, including the current sections 5, 6, and 7.
ANSWER: Thank you for this suggestion. We decided to change the following sections and make better treasability of the reviewed facts including the importance of the packaging system in new-formed sections.
- The types of packaging systems could be discussed in Section 3, thus justifying the biopolymer-based structure selection.
ANSWER: Thank you for this suggestion The types of materials and systems based on them were incorporated in Section 3, with justification of biopolymer-based structures usage.
- The authors are encouraged to summarize and compare the types of biopolymer-based structures and the methods of essential oils incorporation, briefly presenting the processes, advantages, and limitations.
ANSWER: Thank you for this suggestion. Each paragraph that describes a certain type of encapsulation method possesses in the end described advantages and limitations of each structure/method.
- The current Section 8 could become Section 4.
ANSWER: Thank you for this suggestion, we made a new order of sections as we explain in the suggestion no. 1.
- After all suggested changes, they are encouraged to reformulate suitable Conclusions and further perspectives.
ANSWER: The Authors appreciate this suggestion, so we made changes in the last part of our MS, and adjusted it in an appropriate manner.
Round 2
Reviewer 3 Report
The reviewer appreciates the authors' efforts in revising their manuscript according to the previous comments.
However, some major comments and suggestions are still available, as follows:
1. The MS is submitted in the Antibiotics Journal in SI entitled "Advances in the Discovery of Novel Antimicrobial Agents in Nature and Their Applications."
Therefore, the reviewer considers that the Introduction section should begin with the acute problem of microbiological contamination in the food industry, not with plastics and packaging. The authors are encouraged to adapt the Introduction to the scope of the Journal and SI and, in the end, to link it with the food packaging challenges.
2. As an overview of the entire MS, it appears as a long story.
A review as a publication type synthesizes current information in a specific field where the reader wants to find the most critical aspects.
Suggestive figures, tables, etc., touch this scope because the readers are from different domains, not only specialized in food technology.
A. The authors could put the differences between classical and biodegradable food packaging in a table, evidencing the used materials, properties, advantages, and limitations.
B. Suggestive figures/schemes/tables could summarize the main bacteria and yeasts responsible for food contamination.
C. Similar ideas are available for essential oils, constituents, and their mechanisms of antimicrobial action. These aspects are essential because EOs are antimicrobial agents responsible for consuming packaged foods safely.
Lines 414-419 - The authors are invited to check and mention the scientific name of the plant species for all EOs. They are encouraged to check more references and enumerate more essential oils used in the food industry as antimicrobials.
D. The authors could summarize in a table all biopolymers used for biodegradable packaging, grouped in particles, fibers, and gels (line 247), with suitable references. Thus, the MS text could better capture readers' attention and interest.
E. All the methods presented for essential oils incorporation could be summarized in Tables, showing the differences, advantages, and limitations.
F. The authors are encouraged to perform or reproduce with permission suggestive figures to show the most important aspects of the various methods of essential oils incorporation in biodegradable packaging. Thus, the reader could better understand and differentiate the described processes.
All mentioned aspects could be found in numerous other reviews published in various MDPI journals in the same field or other different or similar domains.
Following all these comments and suggestions, the current version could become a valuable, clear, dynamic, and practical review, capturing numerous readers' interest and suitable for publication in the Antibiotics journal.
Author Response
Reviewer 3
The reviewer appreciates the authors' efforts in revising their manuscript according to the previous comments.
However, some major comments and suggestions are still available, as follows:
- The MS is submitted in the Antibiotics Journal in SI entitled "Advances in the Discovery of Novel Antimicrobial Agents in Nature and Their Applications."
Therefore, the reviewer considers that the Introduction section should begin with the acute problem of microbiological contamination in the food industry, not with plastics and packaging. The authors are encouraged to adapt the Introduction to the scope of the Journal and SI and, in the end, to link it with the food packaging challenges.
ANSWER: The Authors would like to thank the Reviewer for a quick and professional review as well as the opportunity to make essential and crucial changes in our work. All the Reviewer’s remarks are accepted and the paper is changed according to their comments. The Authors believe that the changed paper would satisfy the Reviewer' criteria and that it is going to be interesting enough for publishing in the Journal. We decided to revise the manuscript according to the Reviewer' remarks, highlighting the changes directly in the revised manuscript. In accordance with the Reviewer's suggestion, the introduction of the manuscript is fortified to meet the scope of the Journal and SI. We added a part about the acute problem of microbiological contamination in the food industry and linked this part with packaging problems.
- As an overview of the entire MS, it appears as a long story.
A review as a publication type synthesizes current information in a specific field where the reader wants to find the most critical aspects.
Suggestive figures, tables, etc., touch this scope because the readers are from different domains, not only specialized in food technology.
- The authors could put the differences between classical and biodegradable food packaging in a table, evidencing the used materials, properties, advantages, and limitations.
ANSWER: Thank you for this suggestion. The differences, properties, advantages, and imitations are summarized in Table 1 (Section 3), and the text is modified.
- Suggestive figures/schemes/tables could summarize the main bacteria and yeasts responsible for food contamination.
ANSWER: Thank you for your observation. We formed two tables summarizing the most dominant foodborne bacteria, yeasts and fungi and placed them in suitable sections 2.1. and 2.2.
- Similar ideas are available for essential oils, constituents, and their mechanisms of antimicrobial action. These aspects are essential because EOs are antimicrobial agents responsible for consuming packaged foods safely.
ANSWER: We definitively agree that the main constituents and mechanisms of antimicrobial action of the essential oils are essential in understanding the purpose of adding them into packaging systems. Therefore, we summarized all of this trough text in section Incorporation of essential oil as the antimicrobial substance in the biopolymer matrix. On the other hand, we find it unnecessary to additionally form tables or figures related to the same topic, which would just prolonged the manuscript without giving any significant scientific value.
Lines 414-419 - The authors are invited to check and mention the scientific name of the plant species for all EOs. They are encouraged to check more references and enumerate more essential oils used in the food industry as antimicrobials.
ANSWER: Thank you for your observation. We have checked and added the scientific name of all mentioned plant species. Since we gave in this part more than 10 examples of the most used essential oils in the food industry, we found it more than enough. Namely, the use of essential oils in the food industry is on the rise and it is very difficult to mention all of them in this review.
- The authors could summarize in a table all biopolymers used for biodegradable packaging, grouped in particles, fibers, and gels (line 247), with suitable references. Thus, the MS text could better capture readers' attention and interest.
ANSWER: Thank you for this suggestion. Biopolymers are summarized in Table 2.
- All the methods presented for essential oils incorporation could be summarized in Tables, showing the differences, advantages, and limitations.
ANSWER: Thank you for this suggestion. Methods are summarized in Table 3.
- The authors are encouraged to perform or reproduce with permission suggestive figures to show the most important aspects of the various methods of essential oils incorporation in biodegradable packaging. Thus, the reader could better understand and differentiate the described processes.
ANSWER: The most important aspects are clearly highlighted in the three tables, therefore, the incorporation of additional figures or tables will be heavy for the paper leading to the repetition of content in several places in the manuscript. (Section 3).
All mentioned aspects could be found in numerous other reviews published in various MDPI journals in the same field or other different or similar domains.
Following all these comments and suggestions, the current version could become a valuable, clear, dynamic, and practical review, capturing numerous readers' interest and suitable for publication in the Antibiotics journal.
ANSWER: The authors would like to extend our sincere gratitude to the reviewer for the meticulous and insightful assessment of our review. We hope that this time we fulfill the reviewer’s criteria.
Round 3
Reviewer 3 Report
The reviewer appreciates the authors' efforts to increase the MS quality.
Some comments are still available below:
1. Line 205 -Table 2, not Table X. Please check and correct.
2. Line 255 - in Table 3 is better; please check and correct.
3. Table 3 should be revised, placing each used material, property, advantage, or limitation in a separate row for better visualization and understanding.
3. The authors are invited to place Table 4 (line 326) immediately after mentioning it (line 264).
Moreover, the authors are encouraged to complete Table 4 with 2 more columns, advantages and limitations, as mentioned in the table caption.
All Table 4 columns should be suitably named in the first row.
4. The same comment is available for Table 5, mentioned in line 476 and placed in line 513.
The authors are invited to revise Table 5; the current version combines all three incorporation methods classified and described in the MS text (5.1.-5.3. - direct incorporation, emulsification, and liposome formation) and biopolymers nanostructures, cyclodextrins, and nanoclays.
Moreover, they are encouraged to reorganize it, increasing each column size for better visualization.
Is incorporation similar or equivalent to encapsulation (lines 475 and 514)?
5. Based on all evidence, the reviewer still considers that the previous suggestion with some figures/schemes could help clarify all used terms.
6. Line 556 - Desorbtion.
Author Response
Reviewer 3
The reviewer appreciates the authors' efforts to increase the MS quality.
ANSWER: The Authors would like to thank the Reviewer for a quick and professional review as well as the opportunity to make essential and crucial changes in our work. All the Reviewer’s remarks are accepted and the paper is changed according to their comments.
Some comments are still available below:
- Line 205 -Table 2, not Table X. Please check and correct.
ANSWER: Thank you for this observation, we corrected it.
- Line 255 - in Table 3 is better; please check and correct.
ANSWER: Thank you for this observation, we corrected it.
- Table 3 should be revised, placing each used material, property, advantage, or limitation in a separate row for better visualization and understanding.
ANSWER: Thank you for this observation, we corrected it.
- The authors are invited to place Table 4 (line 326) immediately after mentioning it (line 264).
ANSWER: Thank you for this observation, we corrected it.
Moreover, the authors are encouraged to complete Table 4 with 2 more columns, advantages and limitations, as mentioned in the table caption.
ANSWER: Thank you for this observation, but previously table summarized advantages and limitations, so adding this in this table will be only repetation of it. Also, this are disscuss into text, and we believe this may be surplus.
All Table 4 columns should be suitably named in the first row.
ANSWER: Thank you for this observation, we corrected it and made a picture instead of table.
- The same comment is available for Table 5, mentioned in line 476 and placed in line 513.
ANSWER: Thank you for this observation, we corrected it.
The authors are invited to revise Table 5; the current version combines all three incorporation methods classified and described in the MS text (5.1.-5.3. - direct incorporation, emulsification, and liposome formation) and biopolymers nanostructures, cyclodextrins, and nanoclays.
ANSWER: Thank you for this observation, we corrected it.
Moreover, they are encouraged to reorganize it, increasing each column size for better visualization.
ANSWER: Thank you for this observation, we corrected it.
Is incorporation similar or equivalent to encapsulation (lines 475 and 514)?
ANSWER: Thank you for this observation, we changed it in order to avoid confusion.
- Based on all evidence, the reviewer still considers that the previous suggestion with some figures/schemes could help clarify all used terms.
ANSWER: Thank you for this suggestion, we tried once again to make better visualisation of our paper.
- Line 556 - Desorbtion.
ANSWER: Thank you for this observation, we corrected it.